# Optimization of Bioactive Compound Extraction from Saffron Petals Using Ultrasound-Assisted Acidified Ethanol Solvent: Adding Value to Food Waste

**DOI:** 10.3390/foods13040542

**Published:** 2024-02-09

**Authors:** Nikoo Jabbari, Mohammad Goli, Sharifeh Shahi

**Affiliations:** 1Department of Food Science and Technology, Laser and Biophotonics in Biotechnologies Research Center, Isfahan (Khorasgan) Branch, Islamic Azad University, Isfahan 81551-39998, Iran; n.jabbari@khuisf.ac.ir; 2Department of Medical Engineering, Laser and Biophotonics in Biotechnologies Research Center, Isfahan (Khorasgan) Branch, Islamic Azad University, Isfahan 81551-39998, Iran; shahilaser@khuisf.ac.ir

**Keywords:** vacuum-dried saffron petal, bioactive compounds, response surface methodology, acidified-ethanol solvent, ultrasound treatment, optimal extraction conditions

## Abstract

The saffron industry produces large by-products, including petals with potential bioactive compounds, which are cheap and abundant, making them an attractive alternative to expensive stigmas for extracting bioactive components. This study aimed to optimize the extraction conditions of bioactive compounds from vacuum-dried saffron petals using an ultrasound-assisted acidified ethanol solvent. Three factors were considered: ethanol concentration (0–96%), citric acid concentration in the final solvent (0–1%), and ultrasound power (0–400 watt). This study examined the effects of these factors on parameters like maximum antioxidant activity, total anthocyanin content, total phenolic content, and the total flavonoid content of the extraction. This study found that saffron petal extract’s antioxidant activity increases with higher ethanol concentration, citric acid dose, and ultrasound power, but that an increased water content leads to non-antioxidant compounds. Increasing the dosage of citric acid improved the extraction of cyanidin-3-glucoside at different ultrasound power levels. The highest extraction was achieved with 400 watts of ultrasound power and 1% citric acid. Ethanol concentration did not affect anthocyanin extraction. Higher ethanol concentration and greater citric acid concentration doses resulted in the maximum extraction of total phenolic content, with a noticeable drop in extraction at higher purity levels. This study found that increasing the proportion of citric acid in the final solvent did not affect flavonoid extraction at high ethanol concentration levels, and the highest efficiency was observed at 200 watts of ultrasound power. The optimum values of the independent parameters for extracting bioactive compounds from saffron petals included 96% ethanol concentration, 0.67% citric acid concentration, and 216 watts of ultrasound power, resulting in a desirability value of 0.82. This ultrasound-assisted acidified ethanolic extract can be used in the food industry as a natural antioxidant and pigment source.

## 1. Introduction

Studies have shown that medicinal plants contain compounds with significant antioxidant activity, antiradical potential, metal-chelating ability, anticancer properties, and antifungal effects. These compounds, such as phenolics, flavonoids, and carotenoids, can scavenge free radicals, protect against oxidative stress, and potentially prevent chronic diseases. The metal chelating properties of these compounds have been studied for their potential to combat metal-induced toxicity and oxidative stress-related disorders. Additionally, certain compounds derived from medicinal plants have been found to have anticancer properties, exhibiting cytotoxic effects on cancer cells and contributing to apoptosis and cell cycle arrest. Oxidative stress, caused by factors like psychological stress, toxins, industrial lifestyle, infections, drugs, smoking, and obesity, is linked to neurodegenerative disorders like Alzheimer’s and Parkinson’s. Reactive oxygen species (ROS) and free radicals (FR) are linked to various diseases like inflammation, cardiovascular disease, cancer, and diabetes through lipid peroxidation in cellular membranes. To combat these issues, the body needs to provide a constant supply of antioxidants through dietary supplementation. Natural products have been used to prevent and treat many diseases, including cancer, making them potential candidates for developing anti-cancer drugs [1].

The *Crocus sativus*, a perennial plant cultivated in Iran, accounts for 90% of global saffron production. Its flowers, also known as saffron, are valuable spices and medicinal plants with health-enhancing properties for reducing blood pressure and alleviating depression symptoms, as well as antioxidant, antiradical, metal-chelating, anticancer, and antifungal properties [2]. The saffron industry generates large by-products, including large quantities of petals with potential bioactive compounds. Approximately 98.5% of the saffron-flower material is eventually discarded as waste, since only 15 g of spice can be produced from 1 kg of flowers during the production process. This leads to a large number of by-products. Numerous phytochemical components, including flavonoids, anthocyanins, carotenoids, phenolic acids, monoterpenoids, alkaloids, glycosides, and saponins, are present in saffron by-products [3]. These petals are cheap and abundant, making them an attractive alternative to expensive stigmas for extracting bioactive components, especially in the extraction of bioactive components [4].

Bioactive compounds can be recovered from natural products or agro-industrial by-products using conventional methods like Soxhlet, heat reflux, and maceration. However, these methods have low extraction efficiency due to prolonged heat treatment, thermo-sensitive chemical degradation, and significant energy, time, and solvent usage. The grinding procedure has also been used for phenolic compound extraction, but it is ineffective due to high levels of impurities. Therefore, purifying methods are needed to eliminate undesirable components [5]. Innovative extraction technologies like ultrasound, microwave, pulsed electric field, high voltage electrical discharge, supercritical fluid extraction, and pressurized liquid extraction have been developed to protect the stability of bioactive compounds. These techniques offer excellent extraction efficiency, superior final product quality at lower temperatures, quicker kinetics, and reduced energy/solvent consumption. Ultrasounds and other assisted procedures are widely used to maintain sensitive chemicals, offering key benefits such as high extraction efficiency and reduced energy/solvent usage [6]. Acoustic cavitation is a mechanism where sudden bubbles in a liquid grow and collapse, causing high-speed solvent jets to approach the solid surface, thereby accelerating mass transfer and diffusion by increasing the contact surface area between the solid matrix and solvent phase [7,8]. Ultrasound-assisted extraction (UAE) is a green energy-saving technology that offers benefits like time and energy efficiency, lower extraction temperature, less degradation, higher yield, component selectivity, and reproducibility compared to conventional methods [9,10,11]. A study investigated the extraction of lipophilic bioactive compounds from saffron petals using hydroethanolic solvent. Conventional methods were time-consuming, less selective, solvent-intensive, and energy inefficient [5]. According to Ferarsa et al. [5], the cavitation mechanism in ultrasound-assisted extraction consisted of two main steps: (i) dissolving the target com-pounds by causing superficial tissue damage, also known as rinsing or rapid extraction, and (ii) diffusion of the desired chemicals into the extraction medium, also known as slow extraction. When cavitation bubbles collapse at the solid matrix’s surface, the mechanical force causes a breakdown of the plant cell walls, which encourages a rise in the amount of flavonoids during ultrasonication. This enhances mass transfer and increases the solvent–plant material contact surface area [12].

For the first time, this study aimed to optimize the extraction conditions of bioactive compounds from vacuum-dried saffron petals using ultrasound-assisted acidified ethanol solvent as three factors: factor A (ethanol concentration: 0–100% *v*/*v*), factor B (citric acid concentration in the final solvent: 0–1% *w*/*v*), and factor C (ultrasound power: 0–400 watt) using the response surface methodology (RSM), central composite design (CCD) with an alpha of 2, one repeat in axial and factorial points, and three central points. The study investigated the effects of these factors on various parameters, including maximum antioxidant activity (measured by DPPH radical scavenging activity and ferric-reducing antioxidant power assay), total anthocyanin content, total phenolic content, and total flavonoid content of the extraction.

## 2. Materials and Method

### 2.1. Raw Materials and Sample Extract Preparation

The saffron petals were harvested (Vezvan, Isfahan, Iran) in November 2020 and subjected to vacuum-drying (GT 2A, Leybold Heraeus, Koln, Germany) at an absolute pressure of 150 Pa (1.50 mbar). The final solvents recommended by the response surface method were combined with 3 g saffron petal powder in a 1 to 10 ratio. The resulting samples underwent ultrasound-assisted extraction at a temperature of 60 °C, 3 min of extraction time, and a sonication frequency of 28 kHz. The sonication was accomplished with an ultrasonic transducer and a power supply (HAMEG 8150, Mainhausen, Germany). Sonication enhances bioactive compound extraction by facilitating mass transfer from pomace to solvent and improving fluid dynamics along the extraction column [13]. Subsequently, the liquid part of the extract was evaporated under vacuum at 40 °C using an evaporator. The resulting extract was transferred to glass plates and dried until it attained a non-liquid state, while any remaining solvent was drained using a hot water bath maintained at 45 to 50 °C. The dried plates were then sealed, shielded from light with aluminum foil, placed inside a 4-layer plastic cover, and stored in a freezer at −18 °C for further chemical analysis [12]. The experimental study’s entire supply of chemicals came from Merck Co. (Darmstadt, Germany) or Sigma-Aldrich Co. (St. Louis, MO, USA).

### 2.2. Antioxidant Activity Assessment

#### 2.2.1. DPPH Radical Scavenging Activity

One mL of 0.135 mM DPPH (i.e., 2,2-Diphenyl-1-picrylhydrazyl) prepared in methanol was combined with 1.0 mL of aqueous extract containing 0.2–0.8 mg/mL. The reaction mixture was completely vortexed and kept in the dark at room temperature for 30 min. At 517 nm, the absorbance was determined spectrophotometrically. The scavenging ability of the plant extract was calculated as the following [13,14,15,16].
DPPH scavenging activity (%) = [(Abs_control_ − Abs_sample_)/(Abs_control_)] × 100
where Abs_control_ indicates the absorbance of DPPH + methanol and Abs_sample_ indicates the absorbance of DPPH radical + sample (i.e., extract or standard).

#### 2.2.2. Ferric Reducing Antioxidant Power (FRAP) Assay

A 5 µL aqueous extract sample, with concentrations ranging from 10 to 1000 g/mL, was mixed with 180 µL of ferric-TPTZ reagent. The ferric-TPTZ reagent was prepared by combining a 300 mM acetate buffer with pH 3.6, 10 mM TPTZ in 40 mM HCl, and 20 mM FeCl_3_·6H_2_O in a 10:1:1 (*v*/*v*/*v*) ratio. The mixture was then incubated at 37 °C for 5 min. Absorbance at 593 nm was measured using a Thermo Varioskan Flash Microplate Reader (Thermo Scientific, Waltham, MA, USA). The standard curve for FeSO_4_ was linear within the range of 0.15 to 5.00 mM FeSO_4_. The results were reported in terms of FeSO_4_ levels determined using the established standard curves. Each sample was analyzed three times [13,14,15,16].

### 2.3. Total Monomeric Anthocyanin Pigment Determination

The pH-dependent reversible color change of monomeric anthocyanin pigments was observed, where the oxonium form exhibited a colorful appearance at pH 1.0, while the hemiketal form, which was colorless, predominated at pH 4.5. The difference in absorbance of the pigments at 520 nm was found to be associated with the concentration of the pigments. The results were reported in terms of Cyanidin-3-glucoside. It should be noted that the measurements excluded degraded anthocyanins in their polymeric form, which were resistant to color changes at both pH 4.5 and pH 1.0. The absorbance of a test fraction (1.0 mL of aqueous extract containing 0.1–1 mg/mL) diluted with pH 1.0 buffer (0.025 M potassium chloride) and pH 4.5 buffer (0.4 M sodium acetate) was measured at 520 nm and 700 nm. The diluted test samples were compared to a blank cell filled with distilled water. The absorbance measurements were conducted within 20–50 min after preparation. To calculate the concentration of anthocyanin pigments, expressed as Cyanidin-3-glucoside equivalents, the following formula was employed [12,13,15,16]:*Monomeric anthocyanin* (mg⁄kg dry extraction) = (*A* × *MW* × *DF* × 1000)/(*ε* × L)
where A = [(A_520nm_ − A_700nm_) in pH 1.0] − [(A_520nm_ − A_700nm_) in pH 4.5]; MW (molecular weight) = 484.83 g/mol for Cyanidin-3-glucoside; DF = dilution factor; L = path length in cm; ε = 29,000 molar extinction coefficient in L × mol^−1^ × cm^−1^, for Cyanidin-3-glucoside; and 1000 = factor for conversion from g to mg.

### 2.4. Total Phenolic Content (TPC) Determination

The total phenolic content (TPC) of the crude extract was determined using the Folin–Ciocalteu method, employing colorimetric measurement. To prepare the sample, one gram of the extract was diluted with water at a dilution factor of 200. Next, triplicate aliquots of 1.0 mL of the diluted extract were transferred into separate test tubes using a 1 mL transfer pipette. These aliquots were then thoroughly mixed with 5.0 mL of Folin–Ciocalteu reagent, which had been previously diluted 1:10 with distilled water. After shaking for 3 min, 4.0 mL of sodium carbonate solution (7.5% *w*/*v*) was added and mixed thoroughly. The mixtures were then allowed to stand for 30 min in the dark before measuring the absorbance in a single beam UV–vis spectrophotometer (Ocean optics, Orlando, FL, USA) at 765 nm against the blank of methanol pure solvent. On a dry basis, TPC values were expressed as mg gallic acid equivalent (GAE)/kg of extract. To reach the Lambert–Beer linear zone, each solution was diluted with the extraction solvent [13,15,16,17,18].

### 2.5. Total Flavonoid Content Determination

The flavonoid concentration was ascertained using the aluminum chloride colorimetric technique. A total of 1.5 mL of 96% ethanol was added to 500 µL of the extract. Until a final volume of 5 mL was attained, the additional reagents were added in the same manner and quantity as the hydrophilic extracts. After 30 min of darkness for the mixes, the optical density at 415 nm was determined. As a standard, quercetin (QE), a flavanol present in high amounts as an O-glycoside in both fruits and vegetables, was employed. A calibration curve was obtained. The flavonoid content was measured in mg of quercetin per gram of freeze-dried saffron petals [15,16,17,18,19].

### 2.6. Experimental Design and Statistical Analysis

Response surface methodology (RSM) is a statistical approach that uses a series of well-prepared tests to determine the optimal response to multiple causal variables [20]. Independent variables including antioxidant activity-DPPH radical scavenging activity (%), antioxidant activity-FRAP assay (mg Fe^+2^/g vacuum-dried saffron petal), total anthocyanin content (mg Cyanidin-3-glucoside/g vacuum-dried saffron petal), total phenol content (mg Gallic acid/g vacuum-dried saffron petal), and total flavonoid content (mg Quercetin/g vacuum-dried saffron petal) (Table 1). The program Design-Expert 8.1.3 (State-Ease Inc., Minneapolis, MN, USA) was utilized to calculate the linear and quadratic polynomial models coefficients and to perform optimization. Statistical significance was determined based on *p* values less than 0.05. The data was collected in triplicate and averaged to obtain the findings. The regression coefficients (β) were determined by fitting the experimental data to second-order and third-order polynomial models. The response surface analysis employed generalized first and second-order polynomial models.
Y = β_0_ + β_1_A + β_2_B + β_3_C + β_11_A^2^ + β_22_B^2^ + β_33_C^2^ + β_12_AB + β_13_AC + β_23_BC

The coefficients of the polynomial model were represented by β_0_ (i.e., constant coefficient), β_1_, β_2_, and β_3_ (i.e., linear coefficients), β_11_, β_22_, and β_33_ (i.e., quadratic coefficients), and finally β_12_, β_13_, and β_23_ (i.e., interactive coefficients) [21]. The variable A represented the ethanol concentration in percentage, B represented the citric acid concentration in the final solvent as a percentage, and C represented the ultrasound power in watts. The dependent variables or responses, denoted as Y, included the mean values of antioxidant activity based on DPPH and FRAP, total anthocyanin content, total phenol content, and total flavonoid content. Experimental models incorporating linear relationships and quadratic terms were developed based on the acquired data to predict the responses. Subsequently, these models underwent statistical analysis to determine the best-fitting model. The model with the highest R^2^ value was considered a good fit from a statistical perspective. The relative error is a measure of the deviation between the predicted value and the actual value, expressed as a percentage of the true value. It provides information about the quality and reliability of the data obtained [22,23]. To calculate the relative error for each response value, use the following formula:Relative error = [(Predicted value − Actual value)/Actual value] × 100


## 3. Results and Discussion

### 3.1. Fitting the Response Surface Models for Antioxidant Activity Based on *DPPH Radical Scavenging Activity*

ANOVA was used to evaluate the first order polynomial models’ significance (Table 2). For every term in the models, a high F-value and a low *p*-value would suggest a more significant impact on the response variable. The antioxidant activity coefficients and matching *p* values are listed in Table 2. The values of antioxidant activity R^2^, R^2^-adj, adeq accuracy, and coefficients of variation (CV) were, in that order, 0.85, 0.77, 8.53, and 7.76%. The first polynomial model was therefore demonstrated to be more fitting for the antioxidant activity assay than the other models. The ethanol concentration (A), citric acid concentration in the final solvent (B), and ultrasonic power (C) variables did not show significant linear coefficients (*p* > 0.01). The quadratic and cubic coefficients were not significant (*p* > 0.05), while the AB-interactive coefficients were significant (*p* < 0.05). The fit of the model was evaluated using a lack-of-fit test (*p* > 0.05), which shows how well the model can predict variation. The resulting extract exhibits diminishing and rising trends in antioxidant activity at low and high doses of citric acid, respectively, as the purity of ethanol in the final solvent increases. Moreover, increasing the proportion of citric acid in the final solvent demonstrated a reduction in the antioxidant activity of the extract at low and high levels of ethanol concentration, respectively (Figure 1a). The extraction of organic compounds from plant material is closely correlated with the compatibility of solvent ingredients. If the extracted components are well-polarized at the same polarity as the solvent, extraction will be easy; if not, extraction will be challenging. The low water-soluble phenolic content in saffron petals may explain the results, as only models with lower reliability were obtained, and mathematical models of low reliability should not be included. The rate of DPPH free radical scavenging was decreased when saffron petal extraction was performed with more water. This might be explained by the high impurity content of water extraction, which raises the amount of non-phenolic compounds and reduces the antioxidant property [24]. The antioxidant activity of an extract is strongly correlated with the solvent used, due to the individual components’ varying polarity and potential for antioxidant activity. Antioxidant extraction depends critically on the plant materials’ antioxidant compounds’ solubility in the extraction solvent. Plant substances exhibit a correlation between their antioxidant activity and the amount of phenolic chemicals present. Based on the molecular structure of benzoic and cinnamic acids, phenolic acids are free radical receptors. The presence of side chains and phenolic rings in the molecular structure increases the ability to withstand free radical degradation [25]. When compared to phenolic extracts, anthocyanin extracts had the highest antiradical activity across all solvents examined [26]. This suggests that increasing the water-to-alcohol ratio lowers the amount of anthocyanin extracted and, consequently, the antioxidant qualities of the extract. This is in line with the results of our investigation. When compared to non-polar solvents, extraction with highly polar ones produced a high extract yield but a low phenolic and flavonoid concentration. Strong antioxidant molecules are extracted from polar solvents, as indicated by the polarity-dependent increase in overall antioxidant activity [27,28]. Acidic extraction can facilitate the release of phytochemicals by breaking the plant’s cell walls. The performance of extraction depends on the solvent and its concentration. The saffron petal extract obtained from different solvents has different chemical compounds with varying antioxidant activity. Water extraction decreases antioxidant activity with higher citric acid concentration, while alcohol extraction with higher citric acid concentration provides higher activity levels (Figure 1a).

### 3.2. Fitting the Response Surface Models for Antioxidant Activity Based on *FRAP Assay*

The first-order polynomial models’ significance was assessed using ANOVA (Table 2). Table 2 lists the antioxidant activity coefficients and the *p* values that correlate to them. For antioxidant activity, the corresponding R^2^, R^2^-adj, adeq accuracy, and CV values were 0.81, 0.73, 9.30, and 12.33%. Significant linear coefficients were found for the ethanol concentration (A) and ultrasonic power (C) variables (*p* < 0.05). Antioxidant activity was not significantly impacted by the citric acid concentration in the final solvent (B), the AB-, AC-, and BC-interactive terms, or the quadratic and cubic terms of A, B, and C (*p* > 0.05). To assess the model’s fitness, which measures how well it can predict variance, a lack-of-fit test (*p* > 0.05) was run. The study found that the antioxidant activity of the extract from saffron petals increased with increasing ethanol concentration, citric acid dose, and ultrasound power. The antioxidant activity of the extract from saffron petals at all levels of citric acid concentration was observed to increase ultrasound power (Figure 1d) and the purity percentage of ethanol (Figure 1b). This trend was observed at all levels of ultrasound power, with increasing ethanol concentration percentage (Figure 1c) or citric acid dose (Figure 1d). Increased water utilization increases the extraction of compounds that lack antioxidant properties and are thus inappropriate for phenolic compound extraction. High ethanol concentrations in the solvent can modify physical properties such as density, dynamic viscosity, and dielectric constant, resulting in increased antioxidant chemical solubility. The acidified ethanolic extract showed the highest radical scavenging effectiveness. This suggests that its chemical composition varies not just numerically, but also qualitatively. Furthermore, the phenols in the ethanol extract may have been endowed with chemical properties that improve radical scavenging efficacy [29,30]. In general, low water levels, high citric acid percentages, and high ultrasonic power indicated the optimum rate of iron reduction power in this study. Previous research has shown that the extraction method has an impact on the phytochemical content and antioxidant potential of dried saffron petal extracts. Ultrasound-assisted extraction (UAE) with water or low methanol percentages often produced vitamin C and phenolic component results comparable to maceration, allowing for a 50% time reduction in extraction processes. The dried saffron petals had lower levels of total phenolics and anthocyanins but higher levels of antioxidant activity when compared to the spice [31]. The greater power and longer duration of UAE may have contributed to the saffron petal extract’s notable improvement in its antioxidant qualities and phenolic component profile. In general, the acoustic cavitation phenomenon caused by ultrasonic waves results in strong shear stresses inside the structure and has the ability to rupture cell walls, allowing a solvent to enter the plant material and release the intracellular content into the medium [4].

### 3.3. Fitting the Response Surface Models for Total Anthocyanin Content

Table 3 shows the ANOVA used to assess the significance of the first-order polynomial model. A high F-value and a low *p*-value for each term in the models would reflect a more substantial influence on the relevant response variable. Table 3 displays the anthocyanin content coefficient values along with their corresponding *p* values. The anthocyanin content R^2^, R^2^-adj, adeq precision, and CV values were 0.84, 0.78, 12.09, and 6.97%, respectively. As a result, the linear model outperformed the other models in terms of anthocyanin content. The linear coefficient for citric acid concentration in the final solvent (B) and the BC-interactive were both significant (*p* < 0.05). A lack-of-fit test (*p* > 0.05) was used to assess the model’s fitness, which demonstrated the model’s ability to reliably forecast variance. Figure 2a demonstrates that increasing the citric acid dosage enhanced the quantity of cyanidin-3-glucoside extraction at high and low levels of ultrasound power, respectively. The quantity of cyanidin-3-glucoside extracted increased and decreased with increasing ultrasonic power utilized in the extraction process at high and low levels of citric acid consumption in the final solvent. The treatment (400 watt ultrasound power and 1% citric acid concentration) resulted in the greatest quantity of anthocyanin extraction (Figure 2a). According to this study, increasing the purity of ethanol has no effect on anthocyanin extraction and is not a suitable solvent for anthocyanin extraction, while citric acid does increase the rate of anthocyanin extraction to some extent. In the absence of ultrasound, it was suggested that acylated anthocyanins be extracted with a gently acidic solution to prevent hydrolysis. When a moderately acidic medium is used, more and more diverse anthocyanins are extracted. Acid is required to obtain the form of a flavilium cation that is red and stable in acidic solution. Furthermore, at pH = 1.8, anthocyanins have the highest stability [5]. The anthocyanins were distorted by the presence of hydrochloric acid in the solvent, preventing them from hydrolyzing under acidic conditions during solvent evaporation in a vacuum rotary evaporator at 40 °C or deacylation with solvent aliphatic compounds. Increased acidity can improve extraction to some extent, but excessive acidification causes double bond loss and, as a result, anthocyanin loss [32,33]. This is due to the fact that anthocyanin is more stable in acidic solutions and may attach to free radicals in the body such as vitamin C, vitamin E, and beta-carotene. Using acids to aid phytochemical extraction aids in the digestion of plant cell walls, resulting in increased anthocyanin release. The findings agreed with those of a prior work by Li et al. [34], who employed the microwave-assisted extraction of anthocyanins from grape peels with citric acid concentration [28,35]. Ultrasound is a non-thermal food-processing technology that offers an alternative to conventional techniques and can enhance food quality by promoting or damaging enzyme activities at the molecular level. It has been applied to various food by-products, such as fruits, vegetables, and juices, to increase the extraction of polyphenols, improve sensory characteristics, and enhance health benefits due to higher nutraceutical contents. The ultrasonic extraction process of bioactive compounds is a combination of several physical mechanisms such as fragmentation, erosion, the sonocapillary effect, sonoporation, local shear stress, and detexturation [7]. The increase in bioactive components with citric acid treatment could be attributed to its leaching capacity on covalent bonds of the biomass structure, leading to the removal of bioactive compounds from epithelial cells. Further treatment with US could result in the sonolytic cavitation of the biomass structure, improving the availability of phenolic compounds and antioxidant activities. This method has been reported in other fruit by-products, making it an emerging method for agro-allied organizations [36,37]. 

### 3.4. Fitting the Response Surface Models for Total Phenol Content (TPC) 

The ANOVA was used to evaluate the significance of the first-order polynomial model (Table 3). For each term in the models, a high F-value and a lower *p*-value would suggest a more significant influence on the response variable. TPC coefficient values are included in Table 3 along with their matching *p* values. R^2^, R^2^-adj, adeq accuracy, and the CV for the TPC were 0.96, 0.92, 15.63, and 7.02%, respectively (Table 3). Consequently, it was demonstrated that the second-order polynomial model fit the TPC better than the other models. Table 3, Figure 2b shows that the quadratic coefficient of ethanol concentration (A), the linear coefficient for ultrasound (C), the AB-interactive and acid citric in the final solvent (B), and the ultrasound power (C) were all significant (*p* < 0.05). Table 3, Figure 2c,d, and the AC-interactive, BC-interactive, and cubic coefficients were all found to be non-significant (*p* > 0.05). The model’s fitness was assessed using a lack-of-fit test (*p* > 0.05), which showed that the model could correctly predict the variance. Higher and lower ethanol concentrations in the final solvent yielded the maximum extraction of total phenolic content at low and high amounts of citric acid concentration, respectively. There was a noticeable drop in the extraction of the total phenol content at higher ethanol concentration levels and concurrently at greater citric acid doses (Figure 2b). The findings for total phenol content agree exactly with the information in Table 3. Prior studies have demonstrated that anthocyanins from eggplant peels may be extracted more successfully using organic solvents that have undergone acidification [5,13,32]. The extraction success of phenolic compounds is influenced by various factors including the chemical composition, extraction method, sample size, storage conditions, and presence of interfering substances [38].

The phenolic extract, a complex mixture of selectively soluble phenols in various solvents, requires an increase in the solvent’s polarity to enhance its solubility [39]. Despite its great extraction efficiency, the water solvent extracted fewer phenolic compounds than the alcohol solvent, as it is accordance with this study (Figure 2b). Actually, more extractable particles have been dissolved by the water solvent; however, not all of these substances are phenolic. Water as the only solvent can affect the identification and measurement of phenols in mint leaves, affecting total phenol content, flavonoids, and antioxidants. Acetone and 75% ethanol are considered the best solvents for phenol extraction [40,41]. This investigation also found that significant amounts of water had no effect on the extraction of phenolic compounds in comparison to alcohol. Low levels of citric acid concentration enhance phenolic compound extraction at high ethanol concentrations, but excessive concentrations limit the extraction of phenolic compounds. The first molecules to be identified are di-phenhydroquinone and p-benzoquinone, which exhibit a significant acceleration of phenol oxidation at pH values below 4. Consequently, phenols oxidize more quickly at low pH values [42]. Solvent concentration affected both the total phenolic content and the extraction of total monomeric anthocyanins. Because of this, the solvent’s surface tension or dielectric constant may have an impact on how well eggplant peel solubilizes in certain solvents [12]. The US-assisted approach is one of the newly developed strategies of valorization by pertinent agro-allied organizations in light of the improvement in the bioactive yield of fruit waste [37]. Methanolic extracts yielded the highest phenolic compound, followed by water and ethanolic extracts [43]. The number of phenolic compounds increased with increasing citric acid concentration, while extraction using water presented a lower number of compounds. However, when using a lower citric acid concentration, the obtained phenolic compounds were higher. The results were consistent with previous studies on phenolic compounds in plants, which are mainly water-soluble forms. Organic solvents like methanol and ethanol are commonly used for extracting phenolic compounds, with ethanol being the safer option. Antioxidants in plants are typically polar substances, and their activity depends on hydroxyl categories and functional groups [28].

### 3.5. Fitting the Response Surface Models for Total Flavonoid Content (TFC)

ANOVA was used to assess the significance of the second-order, or quadratic, polynomial models (Table 3). For every term in the models, a high F-value and a low *p*-value would suggest a more significant impact on the response variable. The TFC coefficient values and the accompanying *p* values are listed in Table 3. R^2^, R^2^-adj, adeq accuracy, CV, and extraction yield were, in that order, 0.96, 0.93, 18.36, and 4.48%. Consequently, it was demonstrated that the quadratic model outperformed the other models in terms of TFC. Citric acid’s significant linear coefficient (*p* < 0.05) was discovered for the final solvent (B) variable. The cubic coefficient was not significant (*p* > 0.05), but the AB-interactive, quadratic, and citric acid concentration in the final solvent (A) were all significant (*p* < 0.05). The model’s fitness was assessed using a lack-of-fit test (*p* > 0.05), which showed that the model could consistently predict variation. While 96% ethanol exhibited the maximum flavonoid extraction, Figure 3a demonstrates that at the highest levels of ethanol concentration, increasing the proportion of citric acid in the final solvent had no effect on the flavonoid extraction (*p* > 0.05). We saw a considerable increase in flavonoid extraction at lower ethanol concentration levels when the final solvent contained a higher proportion of citric acid. As can be seen in Figure 3b, the extraction process’s ultrasonic power (200 watts) and the end solvent ethanol concentration have a quadratic influence on flavonoid extraction (Table 3). The extraction process’s lowest and maximum levels of ethanol concentration fall within this range. The solvent with the highest flavonoid extraction efficiency was found to be the final one. Figure 3c illustrates that an increasing trend in flavonoid extraction is seen at all ultrasound power levels when the percentage of citric acid concentration in the final solvent increases. At all different levels of citric acid concentration in the final solvent, the highest flavonoid extraction efficiency was observed at 200 watts of ultrasound power. Kaempferols and anthocyanins are the primary chemicals with a high phenolic content, according to the earlier study, which also showed that saffron flower waste includes naturally occurring flavonoid and phenolic compounds. Factors such as origin, altitude, growth circumstances, and picking times can be blamed for variations in these chemicals [44]. This may help to explain why binary solvent systems generate higher yields than those of monosolvent systems. Previous studies have also demonstrated that using organic solvents with acidification improves the extraction of flavonoids from eggplant peels [32,45]. 

Ultrasound power plays a crucial role in the extraction yields of bioactive compounds like total phenolics and flavonoids. It enhances extraction efficiency by creating cavitation in the extraction medium, generating microscopic bubbles that collapse, promoting the release and diffusion of compounds. However, excessive ultrasound power can lead to adverse effects, such as the mechanical degradation or alteration of sensitive compounds like flavonoids. Flavonoids are heat-sensitive and susceptible to degradation under harsh extraction conditions, and the intense energy generated by high ultrasound power can break down or modify their molecular structure, resulting in reduced extraction yields. Additionally, the extraction process is influenced by the solubility of the target compounds in the solvent. The optimal ultrasound power for extraction may vary depending on the compound, solvent, and plant matrix being studied. Researchers conduct preliminary studies to determine the appropriate range of ultrasound power that maximizes extraction efficiency without significant degradation, ensuring the highest overall yield of bioactive compounds while preserving their chemical integrity [18,19].

### 3.6. Optimization of the Solvent Formulation and Survey of Actual and Predicted Data

When the weight and significance values for five responses were determined to be equal, the numerical optimization approach was employed to optimize the extraction conditions (Table 4). In this research, the level of optimization was to be maximized for the antioxidant activity in terms of DPPH radical scavenging activity (18.477–31.152%) and antioxidant activity in terms of ferric-reducing antioxidant power assay (34.496–85.57 mg Fe^+2^/g vacuum-dried saffron petal), total anthocyanin content (3.133–5.444 mg Cyanidin-3-glucoside/g vacuum-dried saffron petal), total phenol content (11.779–28.423 mg Gallic acid/g vacuum-dried saffron petal), and total flavonoid content (41.152–71.856 mg Quercetin/g vacuum-dried saffron petal) of the saffron petal extract samples (Table 4). The best optimal formula for maximum bioactive compound extraction from saffron petals includes ethanol concentration (96%), citric acid concentration in the final solvent (0.67%), and ultrasound power (216 watts) as the predicted results, whose desirability values were equal to 0.82 (Table 4). Furthermore, the disparity between predicted numbers (State-Ease Inc., Minneapolis, MN, USA) and actual (performed in the laboratory) data was minor.

## 4. Conclusions

This study demonstrated that saffron petals, which are cost-effective and abundant by-products of the saffron industry, contain bioactive compounds that can be efficiently extracted using an ultrasound-assisted acidified ethanol solvent. The optimization of extraction conditions, considering factors such as ethanol concentration, citric acid concentration, and ultrasound power, resulted in the highest antioxidant activity, total anthocyanin content, total phenolic content, and total flavonoid content of the saffron petal extract. The optimum values of the independent parameters, i.e., 96% ethanol concentration, 0.67% citric acid concentration, and 216 watts of ultrasound power, yielded a model desirability value of 0.82, as well as antioxidant activity, including DPPH radical scavenging activity (31.15%), antioxidant activity-FRAP assay (74.63 mg Fe^+2^/g vacuum-dried saffron petal), total anthocyanin content (4.61 mg Cyanindin-3-glucoside/g vacuum-dried saffron petal), total phenol content (24.16 mg Gallic acid/g vacuum-dried saffron petal), and total flavonoid content (70.66 mg Quercetin/g vacuum-dried saffron petal). This ultrasound-assisted acidified ethanolic extract holds potential for utilization as a natural source of antioxidants and pigments in the food industry. The minor disparity between predicted and actual data further validates the effectiveness of the optimization approach employed in this research.

## Figures and Tables

**Figure 1 foods-13-00542-f001:**
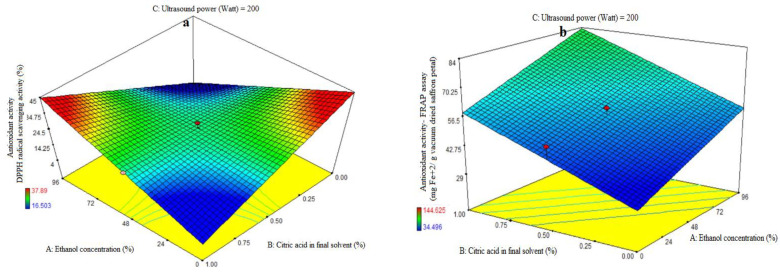
Impact of ethanol concentration (%), citric acid concentration in the final solvent (%), and ultrasound power (watt) on antioxidant activity, as assessed by DPPH radical scavenging activity (%) (**a**) and ferric-reducing antioxidant power (FRAP) (mg Fe^+2^/g vacuum-dried saffron petal) (**b**–**d**) in extracts obtained from the vacuum-dried saffron petals.

**Figure 2 foods-13-00542-f002:**
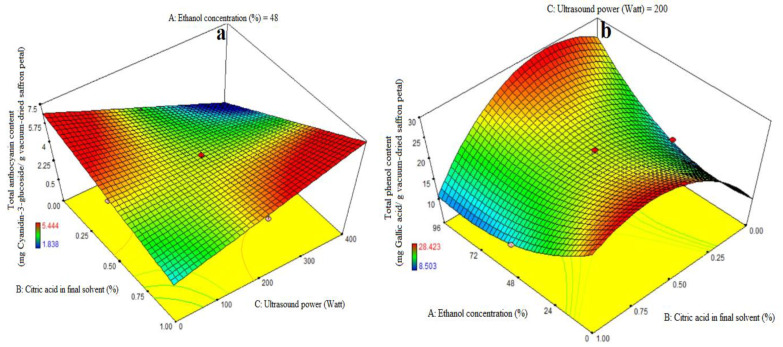
Impact of ethanol concentration (%), citric acid concentration in the final solvent (%), and ultrasound power (watt) on total anthocyanin content (mg Cyanidin-3-glucoside/g vacuum-dried saffron petal) (**a**) and total phenol content (mg Gallic acid/g vacuum-dried saffron petal) (**b**–**d**) in extracts obtained from the vacuum-dried saffron petals.

**Figure 3 foods-13-00542-f003:**
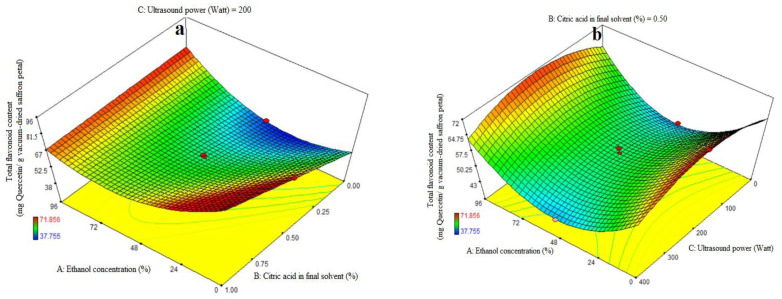
Impact of ethanol concentration (%), citric acid concentration in the final solvent (%), and ultrasound power (watt) on total flavonoid content (mg Quercetin/g vacuum-dried saffron petal). (**a**–**c**) in extracts obtained from the vacuum-dried saffron petals.

**Table 1 foods-13-00542-t001:** The coded and uncoded levels of the independent variables, the composite central design (CCD), and the experimental data for the response surface methodology (RSM) in the extraction of bioactive compounds from vacuum-dried saffron petals.

Independent Variables				Code	Symbol				
					−α	−1	0	1	+α
Ethanol concentration (%)				A	0	24	48	72	96
Citric acid concentration concentration in final solvent (%)				B	0	0.25	0.5	0.75	1
Ultrasound power (watt)				C	0	100	200	300	400
**Run**	**Ethanol Concentration** **(%)**	**Citric Acid Concentration in Final Solvent (%)**	**Ultrasound Power** **(Watt)**	**Antioxidant Activity** **(DPPH-Assay) ***	**Antioxidant Activity (FRAP Assay) ****	**Total Anthocyanin** **Content *****	**Total Phenol** **Content ******	**Total Flavonoid Content *******	
1	72	0.75	300	31.152	85.57	5.444	15.556	55.77	
2	48	0.5	0	34.388	39.965	4.447	21.268	44.322	
3	24	0.25	100	25.149	34.496	4.872	20.696	48.131	
4	48	0.5	200	27.539	106.345	4.038	15.6	49.281	
5	72	0.75	100	27.734	144.625	4.278	22.084	58.203	
6	48	0.5	400	24.208	67.202	4.708	21.901	43.514	
7	24	0.75	100	22.155	54.708	4.528	24.252	61.309	
8	24	0.25	300	18.457	71.961	3.133	11.779	42.18	
9	48	0.5	200	23.695	52.812	4.609	22.127	54.392	
10	48	0	200	37.89	92.541	3.276	13.082	41.152	
11	48	0.5	200	23.096	57.766	3.327	22.876	51.787	
12	72	0.25	300	27.734	77.523	4.812	18.23	51.867	
13	72	0.25	100	20.188	56.691	5.122	23.148	64.063	
14	96	0.5	200	16.503	62.702	1.838	8.503	37.755	
15	48	1	200	24.218	120.996	4.607	12.348	39.521	
16	24	0.75	300	18.477	55.828	5.063	19.506	64.936	
17	0	0.5	200	24.208	50.377	4.554	28.423	71.856	

* DPPH radical scavenging activity (%), ** Ferric reducing antioxidant power (FRAP) assay (mg Fe^+2^/g vacuum-dried saffron petal), *** (mg Cyanidin-3-glucoside/g vacuum-dried saffron petal), **** (mg Gallic acid/g vacuum-dried saffron petal), ***** (mg Quercetin/g vacuum-dried saffron petal).

**Table 2 foods-13-00542-t002:** The analysis of variance (ANOVA) of antioxidant activity in terms of DPPH radical scavenging activity (first-order fitted polynomial model) and antioxidant activity in terms of ferric-reducing antioxidant power (FRAP) assay (first-order fitted polynomial model) in the extraction of bioactive compounds from vacuum-dried saffron petals.

Antioxidant Activity in Terms of DPPH Radical Scavenging Activity (%)							
Source	Coefficient of Final Equation in Terms of Coded Factors	Sum of Squares	df	Mean Square	F-Value	Prob > F	
Model	44.71	103.57	3	34.52	9.58	0.0163	significant
A-Ethanol concentration (%)	−0.406	0.23	1	0.23	0.064	0.8106	
B-Citric acid concentration in final solvent (%)	−41.723	0.43	1	0.43	0.12	0.7438	
AB	0.841	46.71	1	46.71	12.96	0.0155	
Residual	-	18.03	5	3.61	-	-	
Lack of Fit	-	6.4	3	2.13	0.37	0.7884	not significant
Pure Error	-	11.63	2	5.81	-	-	
Cor Total	-	121.59	8	-	-	-	
R^2^: 0.85, Adj-R^2^: 0.77, Adeq precision: 8.53, C.V.: 7.76%							
$\mathrm{Antioxidant}\;\mathrm{activity}\;\left(\mathrm{based}\;\mathrm{on}\;\mathrm{DPPH}\;\mathrm{radical}\;\mathrm{scavenging}\;\mathrm{assay}\right)\;\left(\%\right)=44.71-0.406\;\left(A\right)-41.723\;\left(B\right)+0.841\;\left(\mathrm{AB}\right)$							
**Antioxidant Activity in Terms of FRAP Assay (mg Fe^+2^/g Vacuum-Dried Saffron Petal)**							
**Source**	**Coefficient of** **Final Equation** **in Terms of Coded Factors**	**Sum of Squares**	**df**	**Mean Square**	**F-Value**	**Prob > F**	
Model	17.769	1474.94	3	491.65	10.24	0.0059	significant
A-Ethanol concentration (%)	0.256	471.48	1	471.48	9.82	0.0165	
B-Citric acid concentration in final solvent (%)	28.99	213.79	1	213.79	4.45	0.0728	
C-Ultrasound power (watt)	0.061	395.92	1	395.92	8.25	0.0239	
Residual	-	336.06	7	48.01	-	-	
Lack of Fit	-	323.78	6	53.96	4.4	0.3497	not significant
Pure Error	-	12.27	1	12.27	-	-	
Cor Total	-	1811	10	-	-	-	
R^2^: 0.81, Adj-R^2^: 0.73, Adeq precision: 9.30, C.V.: 12.33%							
$\mathrm{Antioxidant}\;\mathrm{activity}\;\left(\mathrm{based}\;\mathrm{on}\;\mathrm{FRAP}\;\mathrm{assay}\right)=17.769+0.256\left(A\right)+28.99\;\left(B\right)+0.061\left(C\right)$							

**Table 3 foods-13-00542-t003:** The analysis of variance (ANOVA) of the total anthocyanin content (first-order fitted polynomial model), total phenol content (second-order fitted polynomial model), and total flavonoid content (second-order fitted polynomial model) in the extraction of bioactive compounds from vacuum-dried saffron petals.

Total Anthocyanin Content (mg Cyanidin-3-Glucoside/g Vacuum-Dried Saffron Petal)							
Source	Coefficient of Final Equation in Terms of Coded Factors	Sum of Squares	df	Mean Square	F-Value	Prob > F	
Model	6.73	4.5	3	1.5	15.55	0.0007	significant
B-Citric acid concentration in final solvent (%)	−4.022	1.88	1	1.88	19.44	0.0017	
C-Ultrasound power (watt)	−0.016	0.36	1	0.36	3.72	0.0859	
BC	0.027	3.11	1	3.11	32.19	0.0003	
Residual	-	0.87	9	0.097	-	-	
Lack of Fit	-	0.71	8	0.088	0.54	0.789	not significant
Pure Error	-	0.16	1	0.16	-	-	
Cor Total	-	5.37	12	-	-	-	
R^2^: 0.84, Adj-R^2^: 0.78, Adeq precision: 12.09, C.V.: 6.97%							
$\mathrm{Total}\;\mathrm{anthocyanin}\;\mathrm{content}\;\left(\mathrm{mg}\;\mathrm{Cyanidin}-3-\mathrm{glucoside}/g\;\mathrm{freeze}-\mathrm{dried}\;\mathrm{saffron}\;\mathrm{petal}\right)=6.73-4.022\;\left(B\right)-0.016\;\left(C\right)+0.027\left(\mathrm{BC}\right)$							
**Total Phenol Content (mg Gallic Acid/g Vacuum-Dried Saffron Petal)**							
**Source**	**Coefficient of** **Final Equation** **in Terms of Coded Factors**	**Sum of Squares**	**df**	**Mean Square**	**F-Value**	**Prob > F**	
Model	8.71	302.31	7	43.19	22.63	0.0007	significant
A-Ethanol concentration (%)	−0.096	0.67	1	0.67	0.35	0.5749	
B-Citric acid concentration in final solvent (%)	55.69	2.31	1	2.31	1.21	0.3136	
C-Ultrasound power (watt)	0.046	89.9	1	89.9	47.1	0.0005	
AB	−0.313	28.2	1	28.2	14.78	0.0085	
A^2^	0.0028	21.89	1	21.89	11.47	0.0147	
B^2^	−39.15	95.78	1	95.78	50.18	0.0004	
C^2^	−0.0002	33.14	1	33.14	17.36	0.0059	
Residual	-	11.45	6	1.91	-	-	
Lack of Fit	-	11.17	5	2.23	7.97	0.2625	not significant
Pure Error	-	0.28	1	0.28	-	-	
Cor Total	-	313.76	13	-	-	-	
R^2^: 0.96, Adj-R^2^: 0.92, Adeq precision: 15.63, C.V.: 7.02%							
$\mathrm{Total}\;\mathrm{phenol}\;\mathrm{content}\;\left(\mathrm{mg}\;\mathrm{Gallic}\;\mathrm{acid}/g\;\mathrm{freeze}-\mathrm{dried}\;\mathrm{saffron}\;\mathrm{petal}\right)=8.71-0.096\;\left(A\right)+55.69\;\left(B\right)+0.046\;\left(C\right)-0.313\;\left(\mathrm{AB}\right)+0.0028\;{\left(A\right)}^{2}-39.15\;{\left(B\right)}^{2}-0.0002\;{\left(C\right)}^{2}$							
**Total Flavonoid Content (mg Quercetin/g Vacuum-Dried Saffron Petal)**							
**Source**	**Coefficient of** **Final Equation** **in Terms of Coded Factors**	**Sum of Squares**	**df**	**Mean Square**	**F-Value**	**Prob > F**	
Model	39.863	1026.86	6	171.14	30.62	0.0001	significant
A-Ethanol concentration (%)	−0.548	0.28	1	0.28	0.049	0.8304	
B-Citric acid concentration in final solvent (%)	49.048	289.07	1	289.07	51.71	0.0002	
C-Ultrasound power (watt)	0.075	3.39	1	3.39	0.61	0.4619	
AB	−0.565	73.46	1	73.46	13.14	0.0085	
A^2^	0.0088	218.66	1	218.66	39.12	0.0004	
C^2^	−0.0002	81.95	1	81.95	14.66	0.0065	
Residual	-	39.13	7	5.59	-	-	
Lack of Fit	-	26.07	5	5.21	0.8	0.6378	not significant
Pure Error	-	13.06	2	6.53	-	-	
Cor Total	-	1065.99	13	-	-	-	
R^2^: 0.96, Adj-R^2^: 0.93, Adeq precision: 18.36, C.V.: 4.48%							
$\mathrm{Total}\;\mathrm{flavonoid}\;\mathrm{content}\left(\mathrm{mg}\;\mathrm{Quercetin}/g\;\mathrm{freeze}-\mathrm{dried}\;\mathrm{saffron}\;\mathrm{petal}\right)=39.863-0.548\;\left(A\right)+49.048\;\left(B\right)+0.075\;\left(C\right)-0.565\;\left(\mathrm{AB}\right)+0.0088\;{\left(A\right)}^{2}-0.0002\;{\left(C\right)}^{2}$							

**Table 4 foods-13-00542-t004:** Limitations, optimal formula, and data validation for bioactive compound extraction from vacuum-dried saffron petals and antioxidant activity assessment (DPPH radical scavenging and FRAP assay), total anthocyanin content, total phenol content, and total flavonoid content.

Name	Upper	Lower	Upper			
	Goal	Limit	Limit	Weight	Weight	Importance
A-Ethanol concentration (%)	is in range	0	96	1	1	3
B-Citric acid concentration in final solvent (%)	is in range	0	1	1	1	3
C-Ultrasound power (watt)	is in range	0	400	1	1	3
Antioxidant activity-DPPH radical scavenging activity (%)	maximize	18.477	31.152	Y	1	3
Antioxidant activity-FRAP assay (mg Fe^+2^/g vacuum-dried saffron petal)	maximize	34.496	85.57	1	1	3
Total anthocyanin content (mg Cyanidin-3-glucoside/g vacuum-dried saffron petal)	maximize	3.133	5.444	1	Y	3
Total phenol content (mg Gallic acid/g vacuum-dried saffron petal)	maximize	11.779	28.423	1	1	3
Total flavonoid content(mg Quercetin/g vacuum-dried saffron petal)	maximize	41.152	71.856	1	1	3
**Optimal formula for maximum bioactive extraction from saffron petals**						
**Ethanol concentration (96%) and citric acid concentration in final solvent (0.67%) and ultrasound power (216 watts) with a desirability of 82%**						
**Parameters**	**Predicted Value**		**Actual Value**		**Relative Errors (%)**	
Antioxidant activity- DPPH radical scavenging activity (%)	31.152		31.93		2.44	
Antioxidant activity- FRAP assay (mg Fe^+2^/g vacuum-dried saffron petal)	74.629		76.644		2.63	
Total anthocyanin content (mg Cyanidin-3-glucoside/g vacuum-dried saffron petal)	4.613		4.774		3.37	
Total phenol content (mg Gallic acid/g vacuum-dried saffron petal)	24.155		25.17		4.03	
Total flavonoid content (mg Quercetin/g vacuum-dried saffron petal)	70.661		71.721		1.48	

## Data Availability

The data presented in this study are available on request from the corresponding author. The data are not publicly available due to privacy.

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
