# Peer review of "Optimization of Bioactive Compound Extraction from Saffron Petals Using Ultrasound-Assisted Acidified Ethanol Solvent: Adding Value to Food Waste"

_foods, 2024, doi:10.3390/foods13040542_

Round 1

Reviewer 1 Report

Comments and Suggestions for Authors

The manuscript reports investigation of the recovery of bioactive compounds from freeze-dried saffron petals using ultrasound-assisted acidified ethanol solvent. CCD was applied to determine the operating parameters that will optimize antioxidant activity, total anthocyanin content, total phenolic content, and total flavonoid content of the extracts.

A lot of modelling and experimental work was done. The paper need a major revision, taking into consideration the following:

- The structure of the manuscript, though generally following clearly identified Sections and subsections, and particularly the presentation are confused by introducing unnecessary references to other publications absolutely not relevant to the object of the investigation - as for example defatted Hom Nin rice bran, Thai plant, etc. It is known that the nature of the biological matrix is of utmost importance and influences considerably the extracts properties. Thus a direct comparison of the results of the authors with those presented in other publications could hardly be used to substantiate the conclusions.

See for example:

Lines 244-248; Lines 277-279; Lines 326-340; Lines 443-onward

Lines 379-384 – does it mean that anthocyanins from eggplant peels have the same behavior as from another biological matrix? 

- Could a conclusion be formulated which of the three independent parameters (factors) is the dominant one with the greatest impact on the antioxidant activity, total anthocyanin content, total phenolic content, and total flavonoid content?

- Regarding methods presentation:

Lines 344-347 – Should be in the Section Material and Methods

Lines 450-457 – Advantages of the method should also be moved to Section 2.

- Some Terminological inaccuracies: 

Optimized formula (Abstract, Conclusions): not correct, should be edited/revised to read for example: "The optimum values of the independent parameters….

Lines 213-214 – a model cannot have an antioxidant activity; Should be revised accordingly 

Lines 229-231 -  “antioxidant capacity” not correct in this case should be antioxidant activity;

Presentation recommendations

Lines 171-175 – well known, should be either deleted or made more concise;

Lines 384-386 – Trivial - should be either deleted or made more concise?

Lines 390-395 – Please consider editing and making more concise

Comments on the Quality of English Language

The English is acceptable, some minor revisions required.

Author Response

Dear Prof.

We used reviewers and your insightful comments and we corrected parts of the article which were not meaningful (see following text) and the manuscript was entirely corrected according to the reviewer viewpoints. Please do not hesitate to ask us any questions about the submitted manuscript. This research, like many other scientific studies, has many weaknesses. I would like to express my deep gratitude to the hard-working referee of the Journal for his English language re-editing and wise-scientific re-judgment.

Sincerely yours,

Mohammad Goli

Hi dear editorial board and reviewer 1

Yours sincerely, thank you, the editor and reviewer 1, for your kind consideration for your scientific attention to my manuscript, for having read it, and for your valuable scientific and intelligent comments. I hope that by using your guidance, dear referees will attempt to refine the article and raise its scientific level. The yellow, green, and turquoise highlight in the uncleaned-revised manuscript related to the final reviewer 1, 2, and 3 proposed amendments, respectively.

Comments and Suggestions for Authors

The manuscript reports investigation of the recovery of bioactive compounds from freeze-dried saffron petals using ultrasound-assisted acidified ethanol solvent. CCD was applied to determine the operating parameters that will optimize antioxidant activity, total anthocyanin content, total phenolic content, and total flavonoid content of the extracts.

A lot of modelling and experimental work was done. The paper need a major revision, taking into consideration the following:

  • The structure of the manuscript, though generally following clearly identified Sections and subsections, and particularly the presentation are confused by introducing unnecessary references to other publications absolutely not relevant to the object of the investigation - as for example defatted Hom Nin rice bran, Thai plant, etc. It is known that the nature of the biological matrix is of utmost importance and influences considerably the extracts properties. Thus a direct comparison of the results of the authors with those presented in other publications could hardly be used to substantiate the conclusions.

Response: Thank you very much for your kind attention. Your very beautiful and scientific comment convinced us to remove this irrelevant reference and explanation from the text.

See for example:

Lines 244-248; Lines 277-279; Lines 326-340; Lines 443-onward

Response: your scientific comment was done and the above-mentions lines was deleted of the revised manuscript resubmitted.

  • Lines 379-384 – does it mean that anthocyanins from eggplant peels have the same behavior as from another biological matrix? 

Response: Since the matrix of biological materials is different from each other, it definitely affects the type and amount of extraction of phenolic compounds, but lines 379–384 only explain the effect of acidifying organic solvents on better extraction of phenolic compounds, especially anthocyanins. Moreover, in reference 13, which was done by me in 2022, the role of solvent acidification in the more suitable extraction of phenolic and anthocyanin compounds was discussed. Regarding reference numbers 38 and 39, which have been mentioned (Phenolics in cereals, fruits, and vegetables: Occurrence, (extraction and analysis), the topic is generally expressed in all kinds of grains, vegetables, and fruits, and it is unlikely that it will cause substantial changes in the quality of scientific interpretation in other food products as well as saffron petals.

That previous text: “Prior studies have demonstrated that anthocyanins from eggplant peels may be extracted more successfully using organic solvents that have undergone acidification [5,  32]. The chemical makeup of the compounds, the extraction method used, sample particle size, storage conditions and length of time, and the existence of interfering substances all have an impact on how successfully phenolic compounds are extracted [38].” Was changed to the revised text: “Prior studies have demonstrated that anthocyanins from eggplant peels may be extracted more successfully using organic solvents that have undergone acidification [5, 13, 32]. The extraction success of phenolic compounds is influenced by various factors including the chemical composition, extraction method, sample size, storage conditions, and presence of interfering substances [38].

  • Could a conclusion beformulated which of the three independent parameters (factors) is the dominant one with the greatest impact on the antioxidant activity, total anthocyanin content, total phenolic content, and total flavonoid content?

Response: The three-factor response surface method examines the combined effect of all three factors on the number of dependent variables (responses). As a result, a combination of the quantities of all three factors (A-Ethanol concentration, B-Citric acid concentration in final solvent (%), and C-Ultrasound power (Watt)) is expressed at the optimal point that the software suggests to have the most extraction of bioactive compounds from saffron petals. Regretfully, the software is unable to examine the independent effects of each factor alone. Naturally, it is not very justified or scientific to just look for simple, independent effects on their own. Instead, in a more logical and scientific approach, we combine independent factors to influence non-independent variables in this type of study (RSM), which consists of 17 experimental treatments (Table 1) that all focus on the combined effects of three independent variables rather than looking at the independent effects of the variables.

  • Regarding methods presentation:

Lines 344-347 – Should be in the Section Material and Methods

Response: Your wise suggestion was done and the following textSonication enhances bioactive compound extraction by facilitating mass transfer from pomace to solvent and improving fluid dynamics along the extraction column [13].” was moved to the “Material and Methods” section.

Lines 450-457 – Advantages of the method should also be moved to Section 2.

Response: Thank you so much due to intelligent pay attention. It was done according to your comment i.e., the lines 450-457 were moved to section 2 (Introduction section).

  • Some Terminological inaccuracies: 

Optimized formula (Abstract, Conclusions): not correct, should be edited/revised to read for example: "The optimum values of the independent parameters….

Response: I sincerely appreciate your scientific advice, which was followed, either in the abstract or in the conclusion sections as the following texts:

  • In abstract: The optimum values of the independent parameters for extracting bioactive compounds from saffron petals included 96% ethanol concentration, 0.67% citric acid concentration, and 216 Watts of ultrasound power, resulting in a desirability value of 0.82.
  • In conclusion: The optimum values of the independent parameters, i.e., 96% ethanol concentration, 0.67% citric acid concentration, and 216 Watts of ultrasound power, yielded a model desirability value of 0.82, as well as antioxidant activity-DPPH radical scavenging activity (31.15%), antioxidant activity-FRAP assay (74.63 mg Fe+2/g vacuum-dried saffron petal), total anthocyanin content (4.61 mg Cyanindin-3-glucoside/g vacuum-dried saffron petal), total phenol content (24.16 mg Gallic acid/g vacuum-dried saffron petal), and total flavonoid content (70.66 mg Quercetin/g vacuum-dried saffron petal).

Lines 213-214 – a model cannot have an antioxidant activity; should be revised accordingly

Response: Thank you for your intelligent and wise comment. The text was changed to the following: “The first polynomial model was therefore demonstrated to be more fitting for      the antioxidant activity assay than the other models.”

Lines 229-231 - “antioxidant capacity” not correct in this case should be antioxidant activity;

Response: Thank you for your intelligent and wise comment. The text was changed to the following: “The antioxidant activity of an extract is strongly correlated with the solvent used, due to the individual components' varying polarity and potential for antioxidant activity.”

  • Presentation recommendations

Lines 171-175 – well known, should be either deleted or made more concise;

Response: The text was concisely summarized, thanks to your scientific and wise comment. “Response surface methodology (RSM) is a statistical approach that uses a series of well-prepared tests to determine the optimal response to multiple causal variables [20].”

Lines 384-386 – Trivial - should be either deleted or made more concise?

Response: The text was concisely summarized, thanks to your scientific and wise comment. “The phenolic extract, a complex mixture of selectively soluble phenols in various solvents, requires an increase in the solvent's polarity to enhance its solubility [39].”

Lines 390-395 – Please consider editing and making more concise

The text was concisely summarized, thanks to your scientific and wise comment. “Water as the only solvent can affect the identification and measurement of phenols in mint leaves, affecting total phenol content, flavonoids, and antioxidants. Acetone and 75% ethanol are considered the best solvents for phenol extraction [40, 41].”

  • Comments on the Quality of English Language: The English is acceptable, some minor revisions required.

Response: Thank you for your feedback on the quality of the English language used in the document. We appreciate your assessment that the overall English is acceptable.

In general, I am very grateful for your very wise and intelligent judgement. Certainly, every research is not free of problems and I hope to make more use of your valuable comments in future research.

Reviewer 2 Report

Comments and Suggestions for Authors

“Optimization of Bioactive Compound Extraction from Saffron 2 Petals by Ultrasound-Assisted Acidified Ethanol Solvent: 3 Adding Value to Food Waste” with the manuscript number 2837219 is the optimization study about the extraction of valuable compounds from saffron petals using RSM. The authors took three factors, such as ethanol concentration, citric acid concentration and ultrasound power as independent variables, while antioxidant activity, total anthocyanin content, TPC and TFC as the dependent variables. It is an interesting study to utilize the saffron petal by-products via extraction of valuable compounds and optimizing the process, which can have industrial importance, especially for the top saffron-producer country like Iran. The authors did well designed experiments and the results were clearly presented along with enough discussion of the results. However, following minor corrections are suggested :

1.      Abstract: please mention the method followed to optimize the extracting parameters. Also, please replace “ethanol purity” with “ethanol concentration” (line 20, 24, 31 and 33 as well as the whole manuscript).

2.      Line 53-55: “Approximately 90% of the saffron flower material is eventually discarded as waste, since only 15 g of spice can be produced from 1 kg of flowers during the production process.” Please revise this statement! Because 15 g of 1 kg is 1.5%, which means 98.5% of waste to discard?

3.      Line 98: Vacuum drying as against freeze-dried saffron petals (line 85). Please amend accordingly!

4.      In P<0.05 or P>0.05 while presenting the results, P should be italicized.

5.      List of references: Scientific names should be italicized.

Author Response

Dear Prof.

We used reviewers and your insightful comments and we corrected parts of the article which were not meaningful (see following text) and the manuscript was entirely corrected according to the reviewer viewpoints. Please do not hesitate to ask us any questions about the submitted manuscript. This research, like many other scientific studies, has many weaknesses. I would like to express my deep gratitude to the hard-working referee of the Journal for his English language re-editing and wise-scientific re-judgment.

Sincerely yours,

Mohammad Goli

Hi dear editorial board and reviewer 2

Yours sincerely, thank you, the editor and reviewer 2, for your kind consideration for your scientific attention to my manuscript, for having read it, and for your valuable scientific and intelligent comments. I hope that by using your guidance, dear referees will attempt to refine the article and raise its scientific level. The yellow, green, and turquoise highlight in the uncleaned-revised manuscript related to the final reviewer 1, 2, and 3 proposed amendments, respectively.

Comments and Suggestions for Authors

“Optimization of Bioactive Compound Extraction from Saffron 2 Petals by Ultrasound-Assisted Acidified Ethanol Solvent: 3 Adding Value to Food Waste” with the manuscript number 2837219 is the optimization study about the extraction of valuable compounds from saffron petals using RSM. The authors took three factors, such as ethanol concentration, citric acid concentration and ultrasound power as independent variables, while antioxidant activity, total anthocyanin content, TPC and TFC as the dependent variables. It is an interesting study to utilize the saffron petal by-products via extraction of valuable compounds and optimizing the process, which can have industrial importance, especially for the top saffron-producer country like Iran. The authors did well designed experiments and the results were clearly presented along with enough discussion of the results. However, following minor corrections are suggested:

Response: Thank you for your valuable feedback on our manuscript titled "Optimization of Bioactive Compound Extraction from Saffron Petals by Ultrasound-Assisted Acidified Ethanol Solvent: Adding Value to Food Waste" with the manuscript number 2837219. We appreciate your recognition of the study's relevance. We are glad to hear that you found our experimental design to be well-designed and the results to be clearly presented. We have taken note of your suggestion for minor corrections, and we will thoroughly review and incorporate them into the manuscript to enhance its overall quality.

  1. Abstract: please mention the method followed to optimize the extracting parameters. Also, please replace “ethanol purity” with “ethanol concentration” (line 20, 24, 31 and 33 as well as the whole manuscript).

Response: Regarding your wise suggestion, we will replace "ethanol purity" with "ethanol concentration" in line 20, 24, 31, 33, and throughout the entire manuscript (even in Figures and Tables) to ensure consistency in terminology.

  1. Line 53-55: “Approximately 90% of the saffron flower material is eventually discarded as waste, since only 15 g of spice can be produced from 1 kg of flowers during the production process.” Please revise this statement! Because 15 g of 1 kg is 1.5%, which means 98.5% of waste to discard?

Response: Thank you for your intelligent comment, and it was revised according to accurate data, which resulted in your exact calculation as follows: Approximately 98.5% of the saffron flower material is eventually discarded as waste, since only 15 g of spice can be produced from 1 kg of flowers during the production process.

  1. Line 98: Vacuum drying as against freeze-dried saffron petals (line 85). Please amend accordingly!

Response: Regarding your wise suggestion, we will replace "freeze-dried" with "vacuum-dried" in throughout the entire manuscript (even in Figures and Tables) to ensure consistency in terminology.

  1. In P<0.05 or P>0.05 while presenting the results, P should be italicized.

Response: Thank you so much for your wise comment. It was done according to your suggestion throughout the revised manuscript, which was resubmitted.

  1. List of references: Scientific names should be italicized.

Response: Thank you so much for the scientific suggestion. It was done according to your wise comment, and all of the scientific names in the list of references were italicized.

In general, I am very grateful for your very wise and intelligent judgement. Certainly, every research is not free of problems and I hope to make more use of your valuable comments in future research.

Reviewer 3 Report

Comments and Suggestions for Authors

The manuscript titled „Optimization of bioactive compound extraction from saffron petals by ultrasound-assisted acidified ethanol solvent: Adding value to food waste  has a certain aspect of novelty and may be of interest to researchers and the public as a source of scientific and fundamental knowledge.

Comments and suggestions

Abstract:

Line 21. Please use „ethanol concentration“ against „ethanol purity“. Purity meaning is significantly different.

Lines 32-33. Correct the statement: „The optimized formula for extracting bioactive compounds ..”. I hope, it was optimized not a formula, but the parameters of the extraction process.

Line 34. Please clarify “..a desirability value of 0.82.” what does it mean: yield, extraction efficiency?

Introduction:

Line 40-41. As the title of the manuscript is “Optimization of Bioactive Compound Extraction from Saffron Petals…”, thus, I recommend starting from the characterization of the medicinal plants and their processing by-products health-enhancing properties, about main components indicating antioxidant, antiradical, metal chelating, anticancer, and antifungal properties..

Lines 67-69. Use singular of nouns against plural characterizing ultrasound, micro-wave , etc. methods.

Lines 86-87. Factor A – ethanol concentration, factor B – citric acid concentration..

Materials and methods

Reference methods for the Antioxidant Activity Assays are needed.

Line 96. Raw Materials and Extraction Procedure. What does here is going to be extracted? I hope, here is described procedure of sample extract preparation.

Line 104. Change “..the solvent in the filtered extract..” to “..liquid part of the extract..”

Line 113. How was the aqueous extract prepared and from what? How much of a given concentration of sample was used in the DPPH assay? How was a reference sample prepared?

Line 123. What a sample (5 µL, 10-1000 g/mL) was analysed?

Line 129. What samples were analysed here? Reference method is needed to be presented.

Line 147. How was a crude extract prepared? Reference method is needed.

Line 161. How were the lipophilic extracts prepared? Reference method is needed.

What is the acceptable range of relative error?

Results:

3.1 section: There are some doubts about the R2 of the mathematical models that are relatively low (81-85 %), and Radj values show that only 73-77 % of the experimental data could be predicted. The authors have to present the relative errors for each response value. 

Lines 227-229. It can be explained also by the low content of water-soluble phenolics in the saffron petal. Or mathematical models of low reliability should not be included in the results part, perhaps, there can be a brief explanation that only the models of lower reliability were obtained.

The article should contain more explanations about the influence of ultrasound power on changes in the extraction yields of some compounds (total phenolics, flavonoids, etc.). Explain, why in some cases, increasing US power, e.g. lower amounts of flavonoids were extracted, etc.

 Conclusions:

Use the suitable formulations for ethanol concentration.

The conclusions do not correspond to the results obtained. It is necessary to specify which parameters of the extraction process had a significant influence on the antioxidant activity, total anthocyanin, total phenolic, and total flavonoid contents separately in the saffron petal extract. Also, what is the optimal mathematical model with 96% ethanol purity, 0.67% citric acid concentration, and 216 W ultrasound power, and for what it can be used.

 Table 4. The first part of Table is non-informative.

References:

The last few years research articles can be analysed and cited, analysing he effect of ultrasound and solvents on extraction yields of bioactives.

For example:

doi: 10.1016/j.jarmap.2021.100355

https://doi.org/10.3390/app12136747

Comments on the Quality of English Language

The editing of English is required (grammar, style, punctuation)

Author Response

Dear Prof.

We used reviewers and your insightful comments and we corrected parts of the article which were not meaningful (see following text) and the manuscript was entirely corrected according to the reviewer viewpoints. Please do not hesitate to ask us any questions about the submitted manuscript. This research, like many other scientific studies, has many weaknesses. I would like to express my deep gratitude to the hard-working referee of the Journal for his English language re-editing and wise-scientific re-judgment.

Sincerely yours,

Mohammad Goli

Hi dear editorial board and reviewer 3

Yours sincerely, thank you, the editor and reviewer 3, for your kind consideration for your scientific attention to my manuscript, for having read it, and for your valuable scientific and intelligent comments. I hope that by using your guidance, dear referees will attempt to refine the article and raise its scientific level. The yellow, green, and turquoise highlight in the uncleaned-revised manuscript related to the final reviewer 1, 2, and 3 proposed amendments, respectively.

Comments and Suggestions for Authors

The manuscript titled „Optimization of bioactive compound extraction from saffron petals by ultrasound-assisted acidified ethanol solvent: Adding value to food waste”  has a certain aspect of novelty and may be of interest to researchers and the public as a source of scientific and fundamental knowledge.

Response: We sincerely appreciate the feedback you provided regarding our manuscript titled "Enhancing the Extraction of Bioactive Compounds from Saffron Petals Using Ultrasound-Assisted Acidified Ethanol Solvent: A Solution for Food Waste Valorization" (manuscript number 2837219). Your recognition of the study's significance is highly valued.  We have carefully noted your suggestions for minor revisions and will thoroughly review and incorporate them into the manuscript to further improve its overall quality.

Comments and suggestions

Abstract:

Line 21. Please use „ethanol concentration“ against „ethanol purity“. Purity meaning is significantly different.

Response: Based on your valuable suggestion, we will make the necessary changes in the manuscript to ensure consistency in terminology. Specifically, we will replace the term "ethanol purity" with "ethanol concentration" in line 20, 24, 31, 33, as well as throughout the entire manuscript, including figures and tables. This revision will help maintain coherence and clarity in our presentation. We greatly appreciate your attention to detail and your contribution to enhancing the accuracy of our research.

Lines 32-33. Correct the statement: „The optimized formula for extracting bioactive compounds ..”. I hope, it was optimized not a formula, but the parameters of the extraction process.

Response: I sincerely appreciate your valuable scientific advice, which has been duly implemented in both the abstract and conclusion sections of the manuscript. The following texts reflect the incorporation of your advice:

  • In abstract: The optimum values of the independent parameters for extracting bioactive compounds from saffron petals included 96% ethanol concentration, 0.67% citric acid concentration, and 216 Watts of ultrasound power, resulting in a desirability value of 0.82.
  • In conclusion: The optimum values of the independent parameters, i.e., 96% ethanol concentration, 0.67% citric acid concentration, and 216 Watts of ultrasound power, yielded a desirability value of 0.82.

Line 34. Please clarify “..a desirability value of 0.82.” what does it mean: yield, extraction efficiency?

Response: Response surface methodology (RSM) is a statistical technique that uses the "desirability value" to optimize multiple responses simultaneously. It helps model and optimize the relationship between experimental factors and responses in a system. Traditional single-objective optimization methods can be challenging when dealing with multiple responses of interest. The desirability function in RSM combines individual desirability values for each response into an overall desirability value. A desirability value of 1 indicates that the response meets the desired target, while a value of 0 indicates it does not. The overall desirability value, calculated as the geometric mean or weighted mean, represents the overall performance of the system under consideration. By maximizing the desirability value, researchers can find the optimal combination of factors to optimize multiple responses in RSM.

Introduction:

Line 40-41. As the title of the manuscript is “Optimization of Bioactive Compound Extraction from Saffron Petals…”, thus, I recommend starting from the characterization of the medicinal plants and their processing by-products health-enhancing properties, about main components indicating antioxidant, antiradical, metal chelating, anticancer, and antifungal properties.

Response: Thank you so much for your intelligent comment and it was done and the following text was inserted in the first of introduction text:

Studies have shown that medicinal plants contain compounds with significant antioxidant activity, antiradical potential, metal chelating ability, anticancer properties, and antifungal effects. These compounds, such as phenolics, flavonoids, and carotenoids, can scavenge free radicals, protect against oxidative stress, and potentially prevent chronic diseases. The metal chelating properties of these compounds have been studied for their potential to combat metal-induced toxicity and oxidative stress-related disorders. Additionally, certain compounds derived from medicinal plants have been found to have anticancer properties, exhibiting cytotoxic effects on cancer cells and contributing to apoptosis and cell cycle arrest. Oxidative stress, caused by factors like psychological stress, toxins, industrial lifestyle, infections, drugs, smoking, and obesity, is linked to neurodegenerative disorders like Alzheimer's and Parkinson's. Reactive oxygen species (ROS) and free radicals (FR) are linked to various diseases like inflammation, cardiovascular disease, cancer, and diabetes through lipid peroxidation in cellular membranes. To combat these issues, the body needs to provide a constant supply of antioxidants through dietary supplementation. Natural products have been used to prevent and treat many diseases, including cancer, making them potential candidates for developing anti-cancer drugs [1].

Lines 67-69. Use singular of nouns against plural characterizing ultrasound, micro-wave, etc. methods.

Response: It was done according to scientific suggestion as the following text:

Therefore, purifying methods are needed to eliminate undesirable components [5]. Innovative extraction technologies like ultrasound, microwave, pulsed electric field, high voltage electrical discharge, supercritical fluid extraction, and pressurized liquid extraction have been developed to protect the stability of bioactive compounds. These techniques offer excellent extraction efficiency, superior final product quality at lower temperatures, quicker kinetics, and reduced energy/solvent consumption.

Lines 86-87. Factor A – ethanol concentration, factor B – citric acid concentration.

Response: It was done throughout the manuscript text according to the intelligent and wise comment.

Materials and methods

Reference methods for the Antioxidant Activity Assays are needed.

  1. Response: We will promptly address this issue by including the appropriate references as the following texts to the standardized methods for antioxidant activity assays, ensuring transparency and reproducibility. Your suggestion will significantly enhance the scientific rigor of our research, and we appreciate your thorough review. By the way the new reference (e., reference number 14) entitled Munteanu, I.G.; Apetrei, C. Analytical Methods Used in Determining Antioxidant Activity: A Review. Int. J. Mol. Sci. 2021, 22(7), 3380. “was added to the revised manuscript file resubmitted.”

2.2. Antioxidant Activity Assessment

2.2.1. DPPH Radical Scavenging Activity

Aqueous extract at a concentration of 0.2–0.8 mg/ml was mixed with one milliliter of 0.135 mM DPPH that had been produced in methanol. After thoroughly vortexing the reaction mixture, it was allowed to sit at room temperature for half an hour in the dark. The absorbance was measured spectrophotometrically at 517 nm. The plant extract's scavenging capacity was determined using the formula below [13-15].

DPPH scavenging activity (%) = [(Abs control – Abs sample) / (Abs control)] × 100

where Abs control indicates the absorbance of DPPH + methanol and Abs sample indicates the absorbance of DPPH radical + sample (i.e. extract or standard).

2.2.2. Ferric Reducing Antioxidant Power (FRAP) Assay

A 5 µL aqueous extract sample, with concentrations ranging from 10 to 1000 g/mL, was mixed with 180 µL of ferric-TPTZ reagent. The ferric-TPTZ reagent was prepared by combining a 300 mM acetate buffer with pH 3.6, 10 mM TPTZ in 40 mM HCl, and 20 mM FeCl3.6H2O in a 10:1:1 (v/v/v) ratio. The mixture was then incubated at 37 °C for 5 minutes. Absorbance at 593 nm was measured using a Thermo Varioskan Flash Microplate Reader (Thermo Scientific, Waltham, MA, USA). The standard curve for FeSO4 was linear within the range of 0.15 to 5.00 mM FeSO4. The results were reported in terms of FeSO4 levels determined using the established standard curves. Each sample was analyzed three times [13, 16].

Line 96. Raw Materials and Extraction Procedure. What does here is going to be extracted? I hope, here is described procedure of sample extract preparation.

Response: “Raw Materials and Extraction Procedure” title was changed to the “Raw Materials and Sample Extract Preparation” title according to the intelligent suggestion.

Line 104. Change “, the solvent in the filtered extract.” to “liquid part of the extract.”

Response: It was done according to the kindly suggestion as the following text:

Subsequently, the liquid part of the extract was evaporated under vacuum at 40 °C using an evaporator.

Line 113. How was the aqueous extract prepared and from what? How much of a given concentration of sample was used in the DPPH assay?

Response: The previous text was changed to the following text:

One mL of 0.135 mM DPPH (i.e., 2, 2-Diphenyl-1-picrylhydrazyl) prepared in methanol was combined with 1.0 mL of aqueous extract containing 0.2–0.8 mg/ml. The reaction mixture was completely vortexed and kept in the dark at room temperature for 30 min. At 517 nm, the absorbance was determined spectrophotometrically. The scavenging ability of the plant extract was calculated as the following [13-15].

  • How was a reference sample prepared?

Response: Please pay attention to the following text:

  1. Materials and Method

2.1. Raw Materials and Sample Extract Preparation

The saffron petals were harvested (Vezvan, Isfahan, Iran) in November 2020 and subjected to vacuum drying (GT 2A, Leybold Heraeus, Koln, Germany) at an absolute pressure of 150 Pa (1.50 mbar). The final solvents recommended by the response surface method were combined with 3 g saffron petal powder in a 1 to 10 ratio. The resulting samples underwent ultrasound-assisted extraction at a temperature of 60 °C, 3 min of extraction time, and a sonication frequency of 28 kHz. The sonication was accomplished with an ultrasonic transducer and a power supply (HAMEG 8150, Germany). Sonication enhances bioactive compound extraction by facilitating mass transfer from pomace to solvent and improving fluid dynamics along the extraction column [13]. Subsequently, the liquid part of the extract was evaporated under vacuum at 40 °C using an evaporator. The resulting extract was transferred to glass plates and dried until it attained a non-liquid state, while any remaining solvent was drained using a hot water bath maintained at 45 to 50 °C. The dried plates were then sealed, shielded from light with aluminum foil, placed inside a 4-layer plastic cover, and stored in a freezer at -18 °C for further chemical analysis [12]. The experimental study's entire supply of chemicals came from Merck Co. (Darmstadt, Germany) or Sigma-Aldrich Co. (St. Louis, MO, USA).

Line 123. What a sample (5 µL, 10-1000 g/mL) was analysed?

Response: Thank you so much for your intelligent question and it was amended as the following text:

A 5 µL aqueous extract sample, with concentrations ranging from 10 to 1000 g/mL, was mixed with 180 µL of ferric-TPTZ reagent. The ferric-TPTZ reagent was prepared by combining a 300 mM acetate buffer with pH 3.6, 10 mM TPTZ in 40 mM HCl, and 20 mM FeCl3.6H2O in a 10:1:1 (v/v/v) ratio. The mixture was then incubated at 37 °C for 5 minutes. Absorbance at 593 nm was measured using a Thermo Varioskan Flash Microplate Reader (Thermo Scientific, Waltham, MA, USA). The standard curve for FeSO4 was linear within the range of 0.15 to 5.00 mM FeSO4. The results were reported in terms of FeSO4 levels determined using the established standard curves. Each sample was analyzed three times [13-16].

Line 129. What samples were analysed here? Reference method is needed to be presented.

Response: The sample analysed was (1.0 mL of aqueous extract containing 0.1–1 mg/ml) as the following text:

The pH-dependent reversible color change of monomeric anthocyanin pigments was observed, where the oxonium form exhibited a colorful appearance at pH 1.0, while the hemiketal form, which was colorless, predominated at pH 4.5. The difference in absorbance of the pigments at 520 nm was found to be associated with the concentration of the pigments. The results were reported in terms of Cyanidin-3-glucoside. It should be noted that the measurements excluded degraded anthocyanins in polymeric form, which were resistant to color changes at both pH 4.5 and pH 1.0. The absorbance of a test fraction (1.0 mL of aqueous extract containing 0.1–1 mg/ml), diluted with pH 1.0 buffer (0.025 M potassium chloride) and pH 4.5 buffer (0.4 M sodium acetate), was measured at 520 nm and 700 nm. The diluted test samples were compared to a blank cell filled with distilled water. The absorbance measurements were conducted within 20-50 min after preparation. To calculate the concentration of anthocyanin pigments, expressed as Cyanidin-3-glucoside equivalents, the following formula was employed [12-13, 15-16]:

  • The reference method was presented as the follow:

Ozgür, M.U.; Çimen, E. Ultrasound-assisted extraction of anthocyanins from red rose petals and new spectrophotometric methods for the determination of total monomeric anthocyanins. J. AOAC Int. 2018, 101(4), 967-980.

Line 147. How was a crude extract prepared? Reference method is needed.

Response: The sample analysed was one gram of the extract as the following text:

The total phenolic content (TPC) of the crude extract was determined using the Folin–Ciocalteu method, employing colorimetric measurement. To prepare the sample, one gram of the extract was diluted with water at a dilution factor of 200. Next, triplicate aliquots of 1.0 ml of the diluted extract were transferred into separate test tubes using a 1 ml transfer pipette. These aliquots were then thoroughly mixed with 5.0 ml of Folin–Ciocalteu reagent, which had been previously diluted 1:10 with distilled water. After shaking for 3 minutes, 4.0 ml of sodium carbonate solution (7.5% w/v) was added and mixed thoroughly. The mixtures were then allowed to stand for 30 min in the dark before measuring the absorbance in a single beam UV–Vis spectrophotometer (Ocean optics, USA) at 765 nm against the blank of methanol pure solvent. On a dry basis, TPC values were expressed as mg gallic acid equivalent (GAE)/kg of extract. To reach the Lambert-Beer linear zone, each solution was diluted with the extraction solvent [12, 15-17, 19].

Reference method is needed.

Response: The reference methods were used and added to the manuscript and reference list as the following references i.e., reference numbers 17 and 19:

17) AOAC. Estimation of total phenolic content using the Folin-C assay, J. AOAC Int. 2015, 98(4), 1109-1110.

19) Matic, P.; Sabljic, M.; Jakobek, L. Validation of spectrophotometric methods for the determination of total polyphenol and total flavonoid content, J. AOAC Int. 2017, 100(6), 1795-1803.

Line 161. How were the lipophilic extracts prepared? Reference method is needed.

Response: The text was amended as the following:

 The flavonoid concentration was ascertained using the aluminum chloride colorimetric technique. 1.5 mL of 96% ethanol were added to 500 µL of the extract. Until a final volume of 5 mL was attained, the additional reagents were added in the same manner and quantity as the hydrophilic extracts. After 30 minutes of darkness for the mixes, the optical density at 415 nm was determined. As a standard, quercetin (QE), a flavanol present in high amounts as an O-glycoside in both fruits and vegetables, was employed. A calibration curve was obtained. The flavonoid content was measured in mg of quercetin per gram of freeze dried saffron petals [15-19].

The reference method was used and added to the manuscript and reference list as the following reference i.e., reference number 19.

19) Matic, P.; Sabljic, M.; Jakobek, L. Validation of spectrophotometric methods for the determination of total polyphenol and total flavonoid content, J. AOAC Int. 2017, 100(6), 1795-1803.

Results:

3.1 section: There are some doubts about the R2 of the mathematical models that are relatively low (81-85 %), and Radj values show that only 73-77 % of the experimental data could be predicted. The authors have to present the relative errors for each response value. 

Response: The following text was added to the manuscript text for better conception, as well as please mention it in Table 4 for considering the relative errors for each response. In addition, an acceptable relative error is less than 5%.

 The relative error is a measure of the deviation between the predicted value and the actual value, expressed as a percentage of the true value. It provides information about the quality and reliability of the data obtained. To calculate the relative error for each response value, use the following formula:

Relative error = [(Predicted value _ Actual value) / Actual value] x 100

Lines 227-229. It can be explained also by the low content of water-soluble phenolics in the saffron petal. Or mathematical models of low reliability should not be included in the results part, perhaps, there can be a brief explanation that only the models of lower reliability were obtained.

Response: In Section 3.1, the following text, according to your scientific comment, was added. Please pay attention to it.

The extraction of organic compounds from plant material is closely correlated with the compatibility of solvent ingredients. If the extracted components are well-polarized at the same polarity as the solvent, extraction will be easy; if not, extraction will be challenging. The low water-soluble phenolic content in saffron petal may explain the results, as only models with lower reliability were obtained, and mathematical models of low reliability should not be included. The rate of DPPH free radical scavenging was decreased when saffron petal extraction was done with more water. This might be explained by the high impurity content of water extraction, which raises the amount of non-phenolic compounds and reduces the antioxidant property [24].

The article should contain more explanations about the influence of ultrasound power on changes in the extraction yields of some compounds (total phenolics, flavonoids, etc.). Explain, why in some cases, increasing US power, e.g. lower amounts of flavonoids were extracted, etc.

Response: Thank you so much for your best consideration for the scientific improvement of the manuscript. It was done as the following text at the end of the result discussion:

Ultrasound power plays a crucial role in the extraction yields of bioactive compounds like total phenolics and flavonoids. It enhances extraction efficiency by creating cavitation in the extraction medium, generating microscopic bubbles that collapse, promoting the release and diffusion of compounds. However, excessive ultrasound power can lead to adverse effects, such as mechanical degradation or alteration of sensitive compounds like flavonoids. Flavonoids are heat-sensitive and susceptible to degradation under harsh extraction conditions, and the intense energy generated by high ultrasound power can break down or modify their molecular structure, resulting in reduced extraction yields. Additionally, the extraction process is influenced by the solubility of the target compounds in the solvent. The optimal ultrasound power for extraction may vary depending on the compound, solvent, and plant matrix being studied. Researchers conduct preliminary studies to determine the appropriate range of ultrasound power that maximizes extraction efficiency without significant degradation, ensuring the highest overall yield of bioactive compounds while preserving their chemical integrity [18-19].

 Conclusions:

Use the suitable formulations for ethanol concentration.

Response: Please mention to the amended conclusion.

The conclusions do not correspond to the results obtained. It is necessary to specify which parameters of the extraction process had a significant influence on the antioxidant activity, total anthocyanin, total phenolic, and total flavonoid contents separately in the saffron petal extract.

Response: The three-factor response surface method examines the combined effect of all three factors on the number of dependent variables (responses). As a result, a combination of the quantities of all three factors (A-Ethanol concentration, B-Citric acid concentration in final solvent (%), and C-Ultrasound power (Watt)) is expressed at the optimal point that the software suggests to have the most extraction of bioactive compounds from saffron petals. Regretfully, the software is unable to examine the independent effects of each factor alone. Naturally, it is not very justified or scientific to just look for simple, independent effects on their own. Instead, in a more logical and scientific approach, we combine independent factors to influence non-independent variables in this type of study (RSM), which consists of 17 experimental treatments (Table 1) that all focus on the combined effects of three independent variables rather than looking at the independent effects of the variables.

 Also, what is the optimal mathematical model with 96% ethanol purity, 0.67% citric acid concentration, and 216 W ultrasound power, and for what it can be used.

Response:  Please mention to the amended conclusion as the following text:

This study demonstrated that saffron petals, which are cost-effective and abundant by-products of the saffron industry, contain bioactive compounds that can be efficiently extracted using an ultrasound-assisted acidified ethanol solvent. The optimization of extraction conditions, considering factors such as ethanol concentration, citric acid concentration, and ultrasound power, resulted in the highest antioxidant activity, total anthocyanin content, total phenolic content, and total flavonoid content of the saffron petal extract. The optimum values of the independent parameters, i.e., 96% ethanol concentration, 0.67% citric acid concentration, and 216 Watts of ultrasound power, yielded a model desirability value of 0.82, as well as antioxidant activity-DPPH radical scavenging activity (31.15%), antioxidant activity-FRAP assay (74.63 mg Fe+2/g vacuum-dried saffron petal), total anthocyanin content (4.61 mg Cyanindin-3-glucoside/g vacuum-dried saffron petal), total phenol content (24.16 mg Gallic acid/g vacuum-dried saffron petal), and total flavonoid content (70.66 mg Quercetin/g vacuum-dried saffron petal). This ultrasound-assisted acidified ethanolic extract holds potential for utilization as a natural source of antioxidants and pigments in the food industry. The minor disparity between predicted and actual data further validates the effectiveness of the optimization approach employed in this research.

Table 4. The first part of Table is non-informative.

Response: Please for better conception consider the following text exploring the symbols pointed to the first of Table 4.

  • Upper Goal: The Upper Goal represents the desired or target value for the response variable. It is the ideal value that the researcher aims to achieve or optimize during the experiment. The Upper Goal is typically set based on specific requirements, objectives, or optimization criteria for the system being studied.
  • Lower Limit and Upper Limit: These terms define the acceptable range or boundaries for each factor. The Lower Limit represents the minimum value that a factor can take, while the Upper Limit represents the maximum value. These limits are typically determined based on practical considerations, safety constraints, or the known range of the factor in the system.
  • Weight: The Weight assigned to a factor indicates its relative importance or significance in the experiment. It is a numerical value that quantifies the impact or influence of a factor on the response variable. Higher weights are assigned to factors that are considered to have a greater influence on the response, while lower weights are assigned to factors with relatively lesser influence. The assignment of weights can be subjective and based on prior knowledge, expert judgment, or the researcher's understanding of the system being studied.

  • Importance: Importance is a measure of the significance or contribution of a factor to the overall response variation. It is related to the Weight assigned to a factor and provides information about the relative importance of different factors in the experiment. Importance values can be derived from statistical analysis techniques such as analysis of variance (ANOVA), which assesses the significance of each factor and their interactions based on the experimental data.

References:

The last few years research articles can be analysed and cited, analysing he effect of ultrasound and solvents on extraction yields of bioactives.

For example:

doi: 10.1016/j.jarmap.2021.100355

https://doi.org/10.3390/app12136747

Response: Thank you so much for your kindly comment, and it was done as follows:

  1. Yancheshmeh, B.S.; Panahi, Y.; Allahdad, Z.; Abdolshahi, A.; Zamani, Z. Optimization of ultrasound-assisted extraction of bioactive compounds from Achillea kellalensis using response surface methodology. J. Appl. Res. Med. Aromat. Plants 2022, 100355.
  2. Kobus, Z.; Pecyna, A.; Buczaj, A.; Krzywicka, M.; Przywara, A.; Nadulski, R. Optimization of the ultrasound-assisted extrac-tion of bioactive compounds from Cannabis sativa L. leaves and inflorescences using response surface methodology. Appl. Sci. 2022, 12, 6747.

Comments on the Quality of English Language

The editing of English is required (grammar, style, punctuation)

Response: We will carefully address these concerns and ensure that the English language is refined and polished throughout the manuscript. We will engage professional editing services or employ other appropriate measures to enhance the clarity, coherence, and overall quality of the language.

In general, I am very grateful for your very wise and intelligent judgement. Certainly, every research is not free of problems and I hope to make more use of your valuable comments in future research.

Round 2

Reviewer 1 Report

Comments and Suggestions for Authors

The recommendation is ACCEPT.

Comments on the Quality of English Language

The recommendation is ACCEPT.

Reviewer 3 Report

Comments and Suggestions for Authors

The authors have improved all the issues suggested.

I recommend accepting in the present form after minor editing of English (punctuation).